# Performance of non-intrusive uncertainty quantification in the aeroservoelastic simulation of wind turbines

Pietro Bortolotti[1], Helena Canet[1], Carlo L. Bottasso[1], and Jaikumar Loganathan[1,2]

[1]Wind Energy Institute, Technische Universität München, D-85748 Garching, Germany
[2]GE Global Research, Aero Thermal Lab., Hoodi Village, Bangalore, India

**Correspondence:** Carlo L. Bottasso (carlo.bottasso@tum.de)

**Abstract.** The present paper characterizes the performance of non-intrusive uncertainty quantification methods for aeroservoelastic wind turbine analysis. Two different methods are considered, namely non-intrusive polynomial chaos expansion and Kriging. Aleatory uncertainties are associated with the wind inflow characteristics and the blade surface state, on account of soiling and/or erosion, and propagated throughout the aeroservoelastic model of a large conceptual off-shore wind turbine.

Results are compared with a brute-force extensive Monte Carlo sampling, which is used as benchmark. Both methods require at least one order of magnitude less simulations than Monte Carlo, with a slight advantage of Kriging over polynomial chaos expansion. The analysis of the solution space clearly indicates the effects of uncertainties and their couplings, and highlights some possible shortcomings of current mostly deterministic approaches based on safety factors.

## 1 Introduction

The analysis and design of complex engineering systems is typically based on sophisticated numerical models. While in the past these have been mostly based on deterministic formulations, more recently probabilistic approaches have been gaining an increased attention because of their ability to account for uncertainties in both the models and their inputs. Although numerous applications of probabilistic methods can be found in many areas of engineering, so far formal uncertainty quantification has been applied to a lesser degree in the wind energy field. In fact, probabilistic approaches have been used to estimate wind turbine extreme loads, as reported by Dimitrov (2016) and Graf et al. (2018) among others, but comprehensive analyses and design procedures that account for uncertainties have been lagging behind. This can probably be attributed to the inherent complexity of the models describing the behavior of wind turbines and the environment in which they operate. Indeed, wind (and water, in the offshore case) excitations are highly unsteady and characterized by complex phenomena. Additionally, comprehensive wind turbine simulation environments are obtained by coupled multi-physics models, which account for the effects of structural dynamics, aero and hydrodynamics, closed-loop controls, and their mutual interactions. As a consequence of the inherent complexity and computational cost of the resulting simulation tools, most of the analysis and design methods are currently based on deterministic simulation models and uncertainties are, to a large extent, only indirectly accounted for. For example, instead of computing extreme loads from the tails of probability distributions —which would be the probabilistic approach—, artificial deterministic wind time histories are routinely used to generate in a simpler way such limit cases (IEC61400-1, 2005).

The behavior of wind turbines and of the environment in which they operate is profoundly affected by uncertainties. Therefore, time is ripe for investigating rigorous mathematical formulations to evaluate the robustness of designs and to establish confidence levels on outputs of interest. In the literature, already a few authors have taken the first steps in this direction. One of the first wind energy related publications in this field is the paper by Witteveen et al. (2007), where an intrusive formulation of polynomial chaos expansion (IPCE) is used to investigate the effects of uncertainties affecting the Onera dynamic stall model with regard to a 1 MW wind turbine blade. The authors conclude that the model is very sensitive to input uncertainties and that IPCE is able to reconstruct the output statistics with one order of magnitude fewer function evaluations than a standard Monte Carlo (MC) approach. In Petrone et al. (2011), the aerodynamic design optimization of a wind turbine blade is presented, where uncertain levels of contamination affect the airfoil polars along the span of the blade. A Simplex Stochastic Collocation (SSC) method is used for the propagation of the uncertainties, and convergence is compared against the standard MC approach. SSC is found to be significantly more efficient than MC, in the sense that it requires a much smaller number of evaluations of the model for convergence. Multi-objective design solutions are also presented in the same work, investigating trade-offs between maximum power coefficient and minimum sound pressure levels. Another approach for the robust design optimization of wind turbine rotor blades is presented by Campobasso et al. (2016), where uncertainties are assumed in the chord and twist distributions as well as in the prescribed pitch angle. Additional recent efforts in this area have been dedicated to the development of novel stochastic models for the aerodynamic analysis of wind turbine blades (Fluck, 2017).

Modern simulation and design frameworks are typically based on validated comprehensive aeroservoelastic models. Drastic rewritings of such complex codes to incorporate stochastic formulations are clearly undesirable. To enable the use of legacy codes as black boxes within a probabilistic approach, studies have been recently focusing on the augmentation of aeroservoelastic solvers with non-intrusive uncertainty propagation methods. In addition to enabling the reuse of existing software, non-intrusiveness also allows one to rapidly reap the benefits of any modeling improvement, as the problem of uncertainty quantification is essentially decoupled from the details of the underlying simulation model. This approach is followed by Abdallah et al. (2015) using MC. The method, however, is non-intrusive but also typically extremely expensive, because it performs a straightforward exhaustive sampling of the solution space. More sophisticated spectral methods are used in Matthäus et al. (2016) and Murcia et al. (2017). In these three studies, the impact of uncertainties in the soiling of the airfoils and the wind inflow is estimated in terms of the statistics of rotor performance and extreme loads.

The present study expands and refines the work presented in Matthäus et al. (2016), with the primary goal of identifying the most suitable approaches for the propagation of uncertainties throughout aeroservoelastic wind turbine models. A second goal of this work is that of establishing the performance and convergence properties of such methods for this specific application. The in-depth study of uncertainties and their effects on wind turbines is not amongst the goals of this paper, although it is clearly a long term objective of crucial importance. Among the various approaches that are available in the literature (Sudret, 2007), non-intrusive polynomial chaos expansion (NIPCE) and Kriging (Krige, 1951) are considered here, because of their generality and typical good performance on a wide range of different applications.

The study is conducted with reference to a conceptual offshore 10 MW wind turbine, which is representative of the edge of the current technology. The machine is modeled with the code `Cp-Lambda` (Code for Performance, Loads and Aeroelasticity

by Multi-Body Dynamic Analysis), which implements a multi-body formulation for flexible systems with general topologies. The element library includes rigid bodies, non-linear flexible elements, joints, actuators and aerodynamic models (Bottasso et al., 2006; Bauchau, 2011). Uncertainties are assumed both in the wind characteristics, using actual field measurements, and in the aerodynamic properties of the rotor blades, on account of soiling and erosion. Simulations are performed over a range of wind speeds covering the entire operating regime of the machine. The two considered uncertainty propagation methods are compared in terms of their ability to reconstruct the main statistics of key performance indicators and design drivers, including maximum blade tip deflection, ultimate and fatigue loads at various spots on the machine and, finally, annual energy production (AEP). An exhaustive sampling by the classical MC approach is used as benchmark to define the convergence and accuracy of the tested methods. The resulting probabilistic simulation framework can quantify the effects of uncertainties for a comprehensive black-box aeroservoelastic simulator, in support of the analysis and design of wind turbines. This work is an intermediate step towards the inclusion of robust design methods in the procedures described in Bortolotti et al. (2016), which are at present purely deterministic (except than for the standard treatment of wind by the use of multiple realizations of turbulent fields (IEC61400-1, 2005)).

The paper is structured as follows. Section 2 first discusses sources and models of uncertainty for wind turbine aeroservoelasticity, and then briefly presents the two methods considered here for the propagation of such uncertainties. Next, the wind turbine model is presented at the beginning of Sect. 3, followed by a comparison of the convergence trends for the two methods in Sect. 3.2, while an analysis of the results is discussed in Sect. 3.3. Conclusions and recommendations for future work are finally given in Sect. 4.

## 2   Sources of uncertainty and propagation methods

Uncertainties are commonly categorized into two macro families: aleatory and epistemic uncertainties. The former source of uncertainty emerges from the underlying randomness of a process, as for example described by the probability distribution of the wind speed at a certain site. The latter, on the other hand, originates from a lack of knowledge and data. This work considers the effects of aleatory model parameters and inputs with established underlying probability distributions.

Wind turbines are subjected to several sources of uncertainty. In addition to the inherently stochastic character of the wind, which varies in time and space for a multitude of reasons, uncertainties are also present in the aerodynamic characteristics of the machine, in the mechanical properties of the materials, structures and foundations, as well as in the characteristics and performance of many of the sub-systems of a wind turbine. Not only the nominal values of all such parameters are uncertain, but additional sources of uncertainty are introduced by manufacturing processes and the status of wear and tear of each individual machine or component. Additionally, one should not forget that measurements are also uncertain (Tarp-Johansen et al., 2002), so that an absolute real ground truth can not be established in general.

Due to its preliminary character, this study limits its attention to uncertainties affecting the wind inflow and the aerodynamics of the blades. These are typical and relevant examples of aspects of a turbine model that can often only be described in statistical terms, but that also have a profound impact on the behavior and overall performance of the system. It should however be

remarked that the methods analyzed here are general, and in principle applicable to problems other than the ones considered in this work.

## 2.1 Uncertainty in the characterization of the wind

Wind is a natural phenomenon where air particles move dynamically following three-dimensional paths as the result of a number of driving effects. In general, such a complex process can only be measured and described in terms of its statistics. International standards, such as IEC61400-1 (2005), represent wind profiles by a combination of deterministic mean parameters —typically, mean hub-height speed, shear exponent (SE), vertical and horizontal inflow angles— and a turbulence model, which, for an assigned mean turbulence intensity (TI), describes the stochastic variability of the flow field. Each realization of the turbulent wind field is associated with a random seed. By combining the mean flow field with the fluctuations produced by the turbulence model, one obtains a representation of the wind field in space and time. Sufficient durations and number of realizations are typically necessary for the statistics of the generated wind fields to reach convergence.

However, effects such as solar irradiation, seasonal and long term climate changes, vegetation growth and complex terrain conditions play important roles in increasing uncertainties in the characteristics of the wind (Sathe et al., 2011; Ernst and Seume, 2012). These effects may alter in a significant way the statistics of the wind at a given site. All such effects are difficult to measure and quantify with precision, in turn introducing uncertainties in the assumed wind characteristics used for the simulation and design of wind turbines. This is clearly a problem of crucial importance. In fact, for a given turbine and control system, the assumed wind input plays a fundamental role in determining performance and loading, including lifetime and safety.

This work assumes that both TI and SE are uncertain. However, field data often exhibit a correlation between SE and TI that, according to Dimitrov et al. (2015), can be modelled as

$$\mathrm{SE} = \mathrm{SE}_{\mathrm{ref}} + \frac{\mathrm{TI}_{\mathrm{ref}} - \mathrm{TI}}{\mathrm{TI}\,c_{\mathrm{SE}}}. \tag{1}$$

In this expression, $\mathrm{SE}_{\mathrm{ref}}$ is a reference value for the shear exponent, $c_{\mathrm{SE}}$ a correction factor that can be generally assumed equal to 4, and $\mathrm{TI}_{\mathrm{ref}}$ is the value of the turbulence intensity at a wind speed of 15 m/s. Here an uncertain multiplicative factor $k_{\mathrm{TI}}$ is used to perturb an initial distribution of TI over wind speed; when $k_{\mathrm{TI}}$ equals 1, TI at 15 m/s equals $\mathrm{TI}_{\mathrm{ref}}$. Therefore, through Eq. (1), $k_{\mathrm{TI}}$ also introduces a corresponding uncertainty in SE.

Here and in the following all uncertain parameters are modelled with scaled beta distributions. Such distributions are preferred to other possible choices for two reasons: first, they are highly flexible in shaping the probability density function on account of given statistical data and, secondly, they generate bounded distributions with lower and upper limits. This is a necessary feature when modeling parameters that cannot assume negative values. It should be noted, however, that neither NIPCE nor Kriging are bound to scaled beta distributions, and truncated Gaussian, log-normal, uniform distributions or others could also be readily used. The parameters of the beta distribution for the uncertain factor $k_{\mathrm{TI}}$ are reported in Sect. 3.1.

## 2.2  Uncertainty in rotor aerodynamic properties

A second important source of uncertainty in wind turbine simulation and design lies in the aerodynamic characteristics of the rotor. Among other effects, the performance of the airfoils —measured in terms of the aerodynamic coefficients of lift, drag and moment— is considered as a possible major source of uncertainty.

The estimation of airfoil aerodynamic coefficients can be obtained by experimental and numerical techniques. Both approaches are challenging and lead to uncertainties of an aleatory and epistemic nature, especially in the stall and post-stall regimes. Although potentially very significant, such uncertainties are not considered further in this work, which focuses instead on blade surface conditions.

During operation, the surface of a blade may be contaminated by the deposition of dust, dirt, insects and pollen. Additionally, the blade surface can also be altered due to erosion caused by sand and rain. All these effects are typically and particularly prominent at the leading edge, which has a fundamental role in dictating the behavior of airfoils. As a result, changes in surface conditions during operation may result in significant uncertainties in power capture and loading.

Several studies have quantified the impact of erosion and contamination on aerodynamic performance (Khalfallah and Koliub, 2007; Sareen et al., 2014; Zidane et al., 2016). The exact pattern and location of surface changes during operation is a random process, which is largely governed by local effects, such as the local relative speed of the flow with respect to the blade and the local manufacturing surface quality, for example in terms of gel coat thickness and bonding strength (Khalfallah and Koliub, 2007). In the current study, an uncertain level of airfoil profile unevenness is simulated by using the random variable $k_{\mathrm{AF}}$, modeled with a scaled beta probability density function. Variable $k_{\mathrm{AF}}$ is assumed to vary within the values of zero and one, where zero corresponds to the nominal (clean) state of an airfoil, while one corresponds to a contaminated or fully rough state of operation. The airfoil aerodynamic coefficients between these two states are linearly interpolated for any intermediate value of the random variable, as shown in Fig. 1.

Uncertainties in the actual extension of surface degradation along the span of the blade are modelled by introducing a second parameter, termed extent of spanwise degradation (ESD). Parameter ESD is defined as the non-dimensional span length —measured from blade tip— where factor $k_{\mathrm{AF}}$ affects the airfoil coefficients. Since surface degradation typically occurs in the outer portion of the blades, ESD is assumed to follow a beta distribution between zero, which corresponds to a fully clean blade, and 0.5, which implies that the outer 50% of the blade is affected by surface degradation with a severity dictated by $k_{\mathrm{AF}}$.

## 2.3  Methods for uncertainty propagation

As anticipated in Sect. 1, the current literature offers a vast range of methods for the propagation of uncertainties. A detailed overview of the various formulations can be found in Sudret (2007). Among the many options, based on the results presented in Matthäus et al. (2016), the present study considers the regression-based order 3 NIPCE and Universal Kriging (UK), as implemented in DAKOTA (Adas et al., 2015), to propagate the uncertainties discussed in Sects. 2.1 and 2.2.

In Matthäus et al. (2016), the methods of spectral projection and linear regression were tested to determine the polynomial coefficients of NIPCE, the latter typically yielding the best results. In terms of polynomial order, tests were conducted between

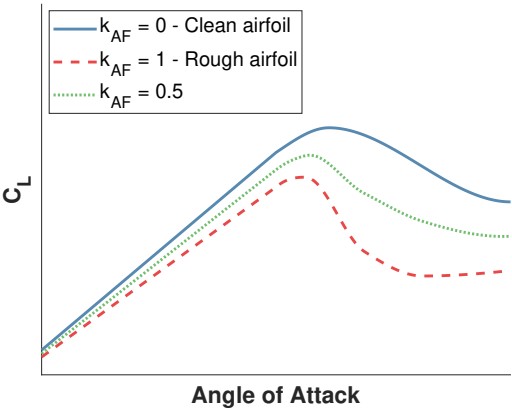

**Figure 1.** Interpolation of the airfoil aerodynamic coefficients between the fully clean and fully rough conditions.

order one and 16. The best results were obtained for order three, while above this value the solution first stopped improving and then deteriorated. It was also found that Universal Kriging is superior to Ordinary Kriging, mostly due to its better adaptability to a general trend in the response.

## 3 Application to a 10 MW wind turbine

5 Here uncertainties in the wind characteristics and in the airfoil polars are propagated throughout the aeroservoelastic model of an offshore wind turbine, with the goal of comparing the performance of the uncertainty quantification methods and of establishing their main convergence characteristics. First, Sect. 3.1 introduces the turbine model together with the assumed uncertainties. Convergence of the statistics is then discussed in Sect. 3.2, while the analysis of the effects of uncertainties on some key outputs is finally presented in Sect. 3.3.

### 10 3.1 Wind turbine model and associated uncertainties

The AVATAR wind turbine is considered in this work, as a representative case of a large offshore wind turbine. This conceptual machine was developed by a consortium of academic and industrial partners within the EU project AVATAR (AVATAR Consortium, 2014-2017), and its main characteristics are summarized in Table 1. In this study, the standard configuration defined by the consortium is used, while the blade inner structure is the one developed at Politecnico di Milano (Croce et al., 2017). 15 Table 2 lists the airfoils used along the span of the blades.

For airfoils DU97-W-300 and DU91-W2-240, which occupy the outermost part of the blade, surface conditions are specified by the two parameters $k_{AF}$ and ESD, by interpolating between fully clean and fully rough aerodynamic coefficients. The clean and rough polars of the two airfoils, which are based on the work performed in the AVATAR project (Méndez et al., 2017), are

**Table 1.** Principal characteristics of the 10 MW AVATAR wind turbine.

| Wind turbine model | 10 MW offshore |
|---|---|
| Wind class | IEC 1A |
| Rated electrical power | 10.0 MW |
| Drivetrain & generator efficiency | 94.0% |
| Rotor diameter $D$ | 205.76 m |
| Hub height $H$ | 127.0 m |
| Nacelle uptilt angle $\Phi$ | 5.0 deg |
| Rotor cone angle $\Xi$ | 2.5 deg |
| Cut-in wind speed $V_{in}$ | 4 m/s |
| Cut-out wind speed $V_{out}$ | 25 m/s |
| Max tip speed $v_{\mathrm{tip_{max}}}$ | 90 m/s |
| Blade mass | 52,874 kg |
| Tower mass | 630.0 ton |

**Table 2.** Spanwise positions of the airfoils.

| Airfoil | Thickness | Position | Airfoil | Thickness | Position |
|---|---|---|---|---|---|
| Circle | 100.0% | 0.0% | DU00-W2-350 | 35.0% | 36.31% |
| Circle | 100.0% | 0.61% | DU97-W-300 | 30.0% | 45.63% |
| DU-600 | 60.0% | 17.00% | DU91-W2-240 | 24.0% | 65.00% |
| DU00-W2-401 | 40.1% | 28.47% | DU91-W2-240 | 24.0% | 100.00% |

reported in Fig. 2. On the other hand, only clean aerodynamic coefficients are used for the airfoils located closer to the blade root, as surface degradation is less likely to happen in this region.

Uncertainties are considered in $k_{\mathrm{TI}}$, $k_{\mathrm{AF}}$ and ESD. As previously explained, the wind parameter SE is not assumed as an independent uncertain variable, but it obeys the relationship of Eq. (1), assuming $\mathrm{SE_{ref}}$ equal to 0.15 and $\mathrm{TI_{ref}}$ equal to 4.9%

5 (see Fig. 3). All uncertainties are assumed to follow the beta distributions whose parameters are reported in Tab. 3. The distribution of turbulence intensity is taken from a measurement campaign conducted in a wind park in the North Sea. The distribution for $k_{\mathrm{TI}} = 1$ is reported in Fig. 3.

An extensive MC is first performed to characterize the solution space. The three uncertainties are propagated throughout the aeroservoelastic model in a power production state at 12 different wind speeds from cut-in to cut-out, considering six turbulent

10 seeds. Eight outputs of interests are analyzed, namely maximum blade tip deflection (MTD), ultimate and damage equivalent load (DEL) of the thrust measured at the main shaft (ThS), ultimate and DEL combined blade root moment (CBRM), ultimate

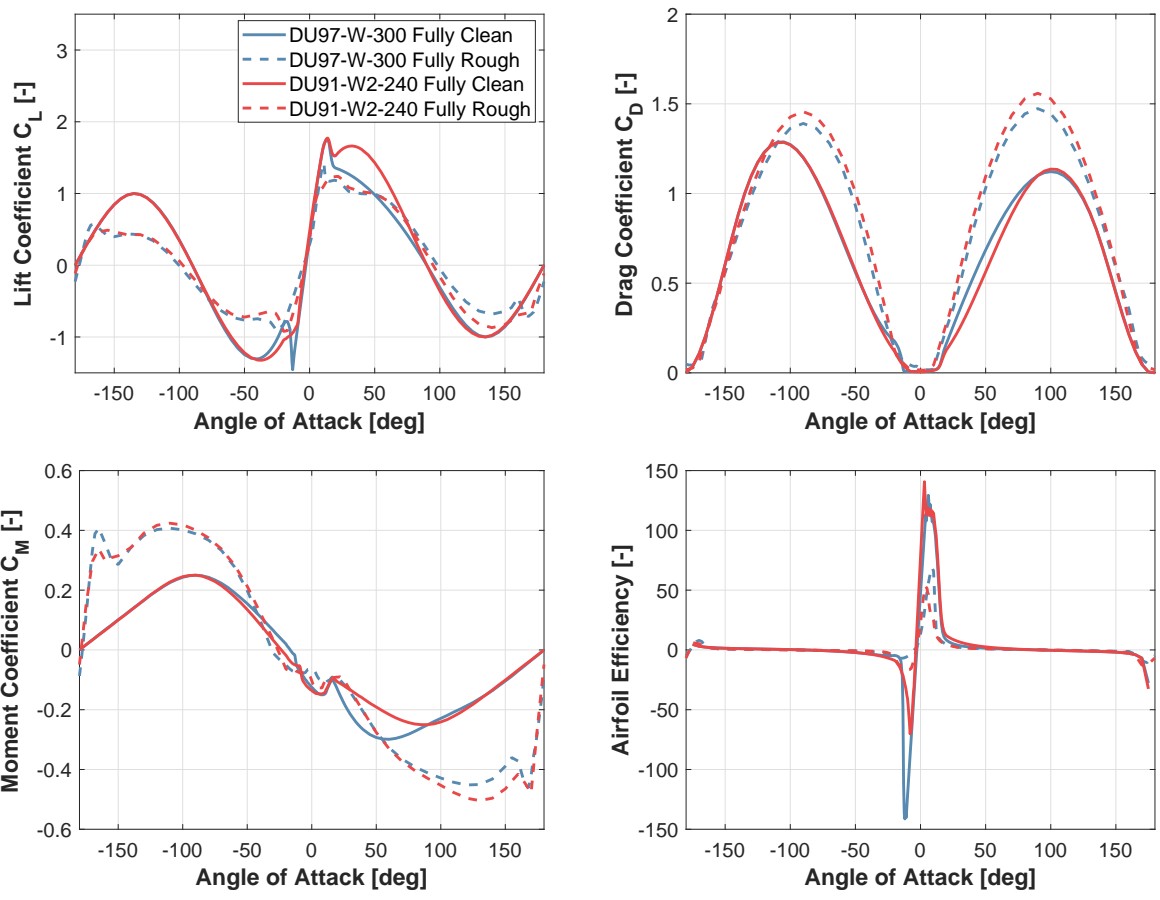

**Figure 2.** Fully clean and fully rough aerodynamic coefficients $C_L$, $C_D$, $C_M$ and airfoil efficiency vs. angle of attack for airfoils DU97-W-300 and DU91-W2-240.

|  | $\alpha$ | $\beta$ | Region |
|---|---|---|---|
| $k_{TI}$ | 3.4 | 6.0 | [0.5 , 2.0] |
| $k_{AF}$ | 2.0 | 6.0 | [0.0 , 1.0] |
| ESD | 2.5 | 4.0 | [0.0 , 0.5] |

**Table 3.** Probability density functions for turbulence intensity factor $k_{TI}$, airfoil roughness $k_{AF}$ and non-dimensional spanwise extent of erosion ESD.

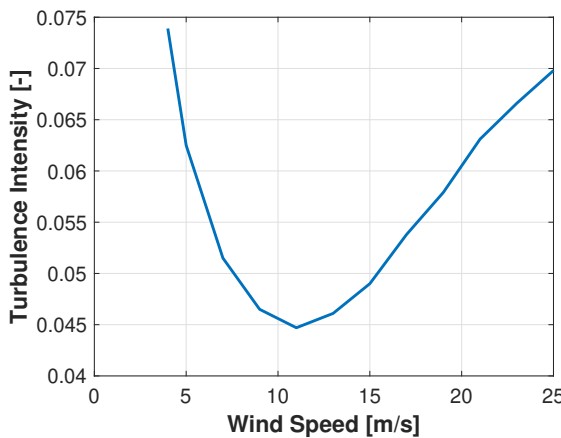

**Figure 3.** Turbulence intensity distribution for varying wind speed.

and DEL combined tower base moment (CTBM), and finally annual energy production (AEP). MTD and ultimate ThS, CBRM, and CTBM are obtained by computing the maximum overall value across all time steps and wind speeds. DELs and AEP are instead averaged via the Weibull distribution corresponding to wind class IA, which is characterized by a shape factor of 2 and an average wind speed at hub height of 10 m/s (IEC61400-1, 2005).

The MC analysis was stopped at 1,100 evaluations, where the convergence of mean and standard deviations for all quantities consistently returned variations below 1% of their average values. While convergence is rapidly obtained for the mean values of the eight outputs of interest, standard deviations require a significantly higher number of evaluations to reach convergence. The statistics of the outputs are reported in Table 4.

Here, six seeds were used to limit the computational cost of the MC analysis, following accepted international standards
(IEC61400-1, 2005). However, as reported in the literature (Dimitrov et al., 2015; Graf et al., 2018), this number might not always be adequate. This is confirmed also here, as the use of only six seeds does not guarantee the full convergence of all quantities, especially in terms of standard deviations, as shown by Fig. 4. While the differences in AEP and DELs are indeed small, this is not true for the ultimate loads. A better understanding of the convergence of results with the number of turbulent realizations should be the subject of future work, as discussed in Sect. 4.

**3.2   Convergence analysis**

The convergence of the uncertainty propagation methods is studied first. The analysis considers mean and standard deviation of AEP, maximum tip displacement, thrust, combined blade root moment, combined tower base moment and the corresponding damage equivalent loads.

Order-three NIPCE and UK, both as implemented in Dakota (Adas et al., 2015), are tested against the MC benchmark
presented in Sect. 3.1. To ensure a fair comparison, a MC sampling strategy is adopted for both NIPCE and Kriging. The number of training data samples follows the relation $R = r\,N_t$, where $r$ is the collocation ratio, varying from 0.6 to 8, and $N_t$ is

**Table 4.** Main statistics of the eight outputs of interest for 1,100 MC function evaluations. MTD: maximum tip deflection; ThS: thrust at main shaft; CBRM: combined blade root moment; CTBM: combined tower base moment; DEL: damage equivalent load; AEP: annual energy production.

|          | Mean        | Standard deviation | Coefficient of variation |
|----------|-------------|--------------------|--------------------------|
| MTD      | 6.99 m      | 0.11 m             | 1.58 %                   |
| ThS      | 2.08 MN     | 0.02 MN            | 1.02 %                   |
| DEL ThS  | 0.34 MN     | 0.05 MN            | 13.79%                   |
| CBRM     | 56.29 MNm   | 0.63 MNm           | 1.12 %                   |
| DEL CBRM | 29.51 MNm   | 2.61 MNm           | 8.83 %                   |
| CTBM     | 236.05 MNm  | 2.20 MNm           | 0.93 %                   |
| DEL CTBM | 46.79 MNm   | 7.82 MNm           | 16.72 %                  |
| AEP      | 53.71 GWh   | 0.29 GWh           | 0.54 %                   |

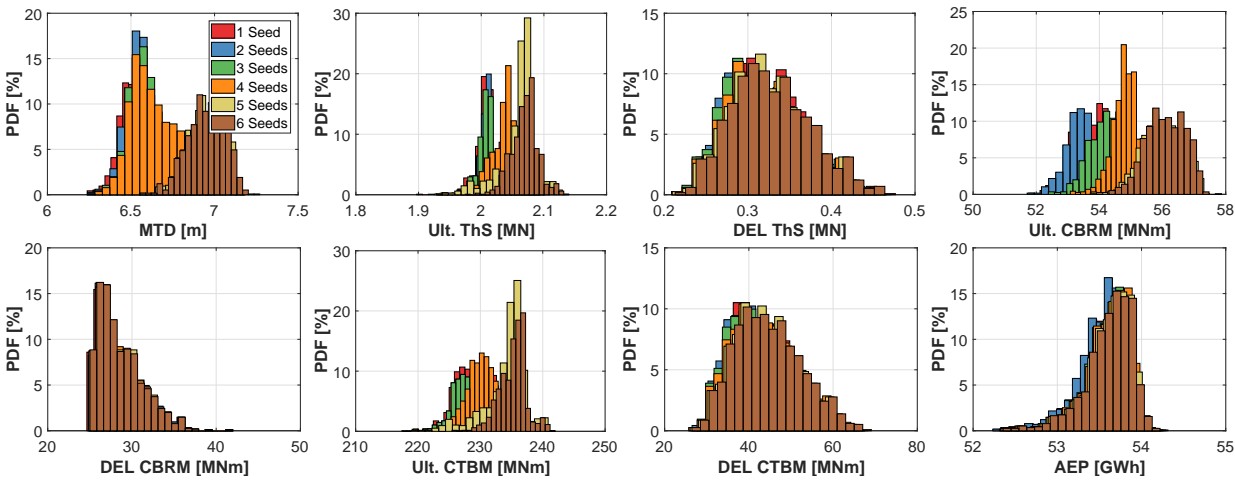

**Figure 4.** Probability density functions of key output metrics for varying number of seeds. Each case is based on 1,100 sampling points.

the total number of terms considering a total-order expansion. The collocation ratio is defined as the ratio between the number of function evaluations used to train the model and the total number of terms in the chaos expansion. On the resulting response surface, an extensive MC sampling with $100,000$ points is conducted to extract mean and standard deviation.

Both NIPCE and UK appear to be capable of estimating the eight outputs of interest at a much reduced number of function evaluations compared to MC. In addition, UK consistently converges faster than the other two methods, with a reduction of one-two orders of magnitude with respect to MC for the estimation of the output mean and standard deviation. The plots reported in Fig. 5 provide for a visualization of these results. In the figure, a gray area represents the 95% confidence intervals for the finite (here equal to 1,100) number of sampling points used in the MC analysis. The grey band could be made narrower by increasing the number of samples.

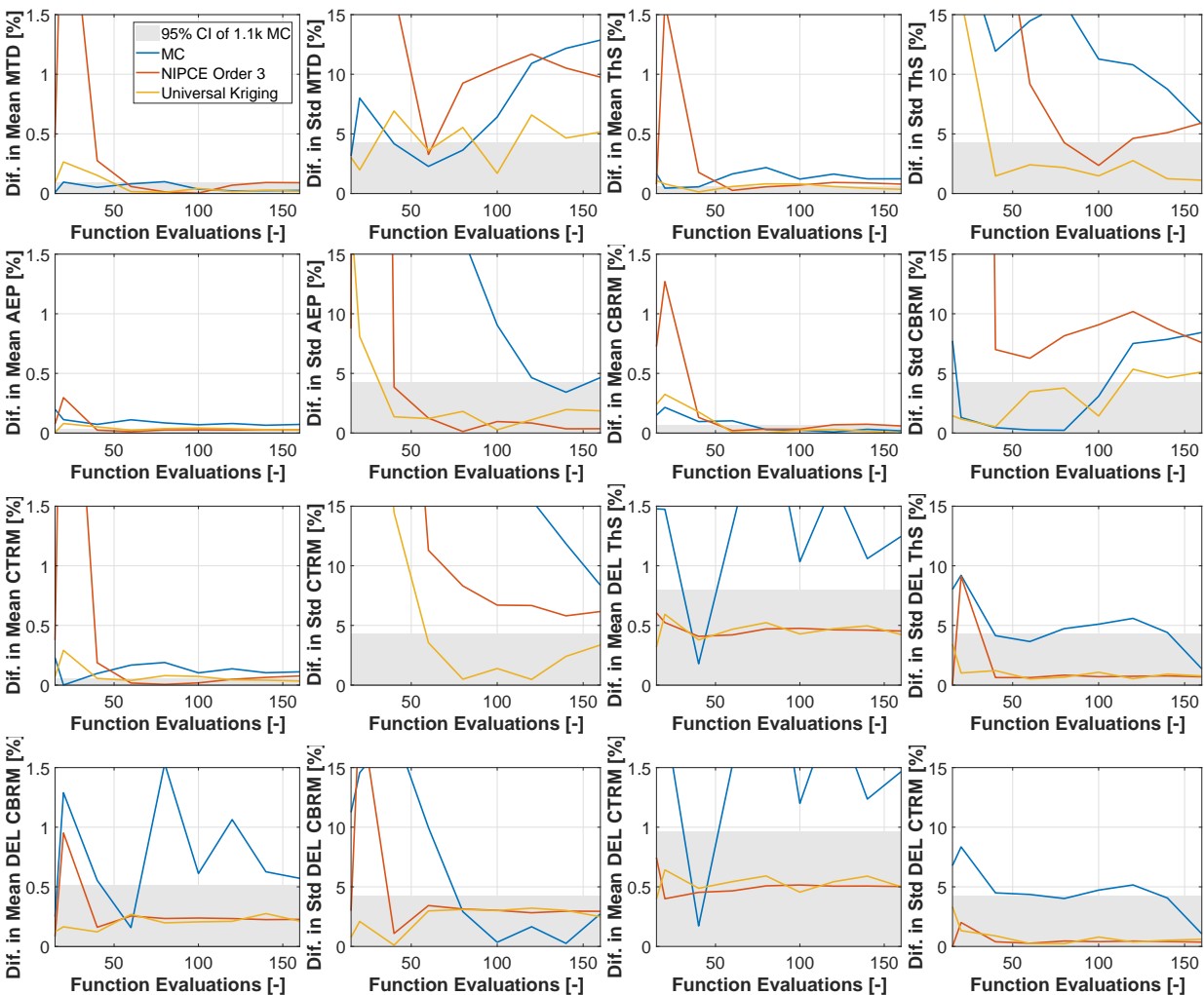

**Figure 5.** Convergence of mean and standard deviation for key output quantities. The gray area reflects the potential inexactness of the MC benchmark, and it represents the 95% confidence intervals for 1,100 sampling points.

### 3.3 Effects of uncertainties on outputs of interest

The results obtained by UK with 40 function evaluations are then subjected to a more detailed analysis. Response surfaces for the eight outputs of interest and their corresponding probability density functions are shown in Fig. 6. The plots are generated by first training the UK model with 40 points and then evaluating it with a random sample of 1 million points. Given the three-dimensionality of the solution space, two-dimensional surfaces are plotted for a constant $k_{TI}$ equal to one.

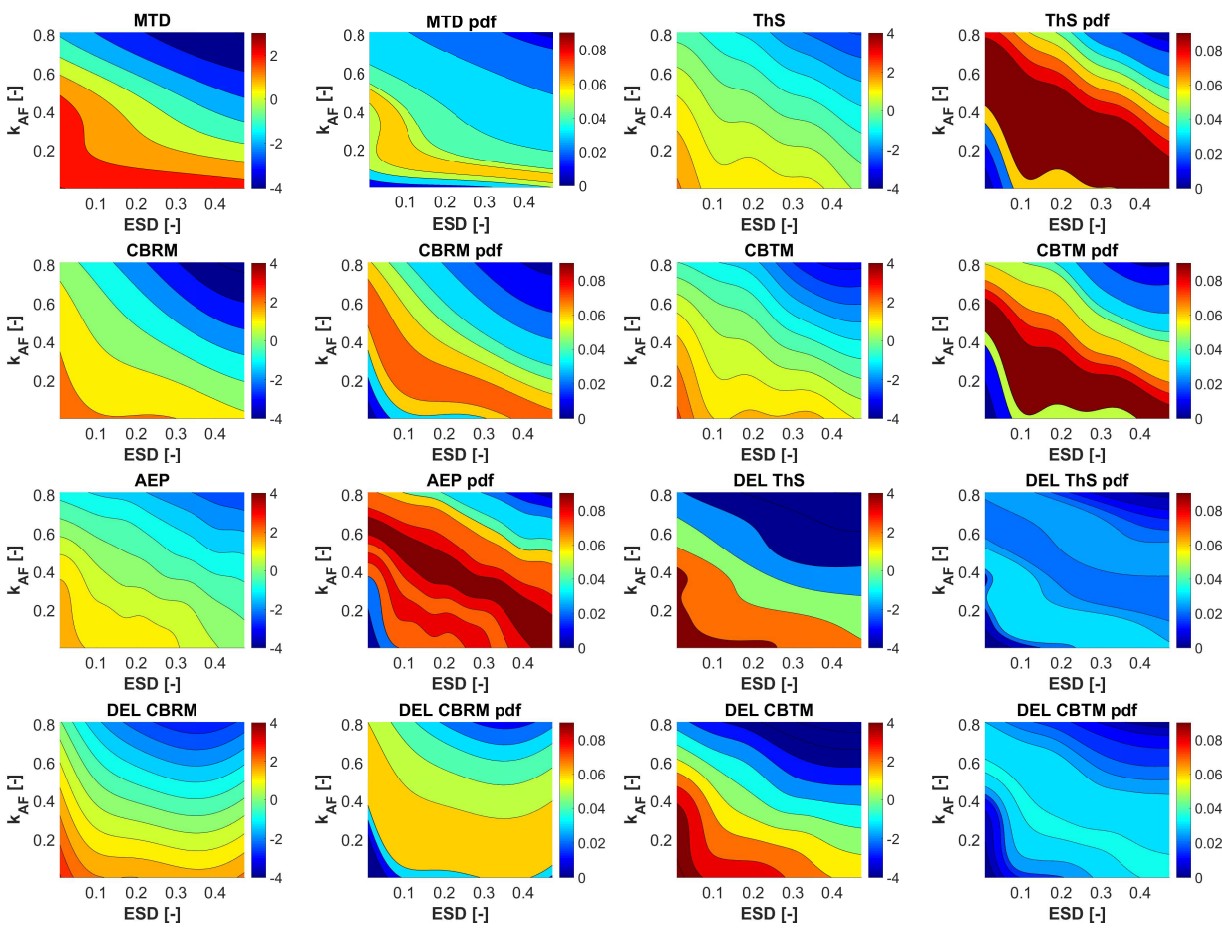

**Figure 6.** Key outputs (in percent difference with respect to the mean value) and corresponding probability density functions, for $k_{TI}$ equal to one.

The contour plots visibly show non linearities. Additionally, they also show that the condition corresponding to a fully clean rotor, namely ESD and $k_{AF}$ equal to 0 (bottom left corner of each plot in Fig. 6), generates the highest values for all eight outputs of interest (left plots). However, according to the input distributions of Table 3, these conditions also have a very low probability of occurrence (right plots). For MTD and the three key loads ThS, CBRM and CTBM, this means that the deterministic simulations prescribed by the standards overestimate the actual output values. Since the variations in the outputs

are limited, and typically in the range of ±3%, these results might appear to suggest that the conventional safety factors equal to 1.2 or 1.3 may be excessive. It is however clear that this analysis is purely limited to the effects of surface roughness and some wind inflow parameters, and a more comprehensive analysis should be conducted before drawing any final conclusion or recommendation. It should also be remarked that the non-intrusive uncertainty propagation methods used here would indeed

allow for such a more general analysis in a rather straightforward manner.

MTD provides for an interesting example. International standards prescribe MTD to be 30% lower than tower clearance. The top left plots in Fig. 6 show that the largest probability of occurance corresponds to MTD values that fall within ±1% of the mean, while very low probabilities are associated to the value of MTD obtained in the deterministic condition prescribed by the standards ($k_{\mathrm{AF}}$ and ESD equal to 0). Similarly, a deterministic analysis overestimates AEP by about 3%, while the uncertainty

analysis shows an equal probability within a range of ±1.5% from the mean value.

In addition, the contour plots of MTD and AEP indicate a fairly linear behavior of the solution space, where the two outputs show a maximum variation along the 45 degree bi-sector. This follows from the fact that, as expected, the rotor is more loaded for clean airfoils and a low extent of erosion (both $k_{\mathrm{AF}}$ and ESD equal to zero), generating higher AEP and MTD. These variations are apparently approximately linear, and as a result the region of maximum probability aligns with the expected

values of $k_{\mathrm{AF}}$ and ESD.

## 4   Conclusions and outlook

This work has reported on the first steps towards the development of a framework for the non-intrusive propagation of uncertainties throughout black-box aeroservoelastic wind turbine models. Non-intrusiveness is key to the reusability of legacy models, and for rapidly reaping the benefits of modeling improvements without the need for a deep rewriting of such complex

codes.

NIPCE and UK were applied to a large state-of-the-art conceptual wind turbine, considering both power capture, tip deflection and some typical design-driving loads as performance indicators. Uncertainties were considered in both the wind inflow conditions and the roughness of the blades, on account of soiling and/or erosion. For both methods, comparisons to standard brute-force Monte Carlo predictions indicate a good performance in terms of quality at a significantly lower computational

cost. Of the two, UK appears to consistently converge faster than NIPCE.

The analysis of the results indicates non-linearities and couplings among the various sources of uncertainty. In addition, it was found that the deterministic conditions prescribed by international design standards generate maximum values of loads and power production, which however are typically associated with a very low probability of occurrence. Although the results obtained here are not comprehensive enough to draw any significant conclusions, they do suggest that the use of formal

mathematically-based methods of uncertainty propagation may lead to a revision of typical safety factors, in the interest of more cost-competitive —but still fully safe— designs.

The present study should be refined in several important aspects. To start, the problem of turbulent realizations deserves specific attention. Here the number of turbulent seeds typically recommended by design standards was used, but appeared

not to be always sufficient for guaranteeing convergence of the statistics. If the number of seeds needs to be increased in a substantial manner to ensure convergence, this might require a change in the methodological approach, as the computational cost might become prohibitive. In this sense, the use of surrogate models, instead of the high-fidelity ones used here, might become attractive. An additional problem of interest is the computation of extreme states, which populate the tails of the

probability distributions and often act as design drivers. Here, ad hoc sampling strategies have been developed by the statistical research community, and could be applied to the problem at hand (Graf et al., 2018). Other sophisticated sampling methods, such as Latin Hypercube Sampling or Hammersley Sampling (Hosder et al., 2007; Eldred et al., 2009), have been described in the literature and will be topic of future studies. Furthermore, additional sources of uncertainty should be investigated. In fact, in principle many parameters and inputs can be assumed to be uncertain. However, a comprehensive knowledge of the

role played by the various uncertainties and their couplings is still largely missing. A ranking of uncertainties and a deeper understanding of their effects is a very worthwhile endeavour, which might have a significant role in the future design of wind energy systems.

*Acknowledgements.* The authors wish to acknowledge A. Croce and L. Sartori of the Department of Aerospace Science and Technology of Politecnico di Milano for providing the data of the 10 MW AVATAR wind turbine. Additionally, credit goes to D. von Terzi and T. Maeder

of GE Global Research for the fruitful discussions and the partial financial support of this research.

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
