# Peer review of "Performance of non-intrusive uncertainty quantification in the aeroservoelastic simulation of wind turbines"

_Wind Energy Science, 2018_

## Referee Comment (RC1) · Anonymous Referee #1 · 17 Feb 2019

The authors present a well-written and well-motivated application of standard non-intrusive uncertainty calculations to the estimation of loads on a wind turbine.

There are, however, a few areas where additional clarity or corrections to the text and figures are required:

Abstract: In the first line, "uncertainties" should be replaced by "aleatory uncertainties". The last sentence should also be made more specific as to what the effects and shortcomings are.

Section 2, page 1, line 21: "uncertainties are [...] only indirectly accounted for" - the concept of "indirect" uncertainty calculation should be explained, preferably with a ci-

tation of an example. Section 2.1, page 4, line 14: "give" should read "given". Section 2.1, page 4, lines 16-20: The choice of a Beta distribution (actually strictly speaking a scaled Beta distribution, since the input values do not always lie between 0 and 1 - see also Section 2.2, page 5, line 21 for ESD) is not sufficiently motivated. This distribution has some specific purposes in the statistical literature, in particular for expressing an uncertainty distribution over a probability. The reason for the turbulence intensity to be modified by a factor which lies between 0.5 and 2 is not explained, since it implies that the turbulence intensity corresponding to $k\_TI=1$ will not actually be the mean or median of this distribution? A log-normal or truncated Gaussian distribution (with mean or median set to 1) would appear more appropriate. Section 2.1, page 4, lines 21-22: The Dimitrov paper does not appear to contain this equation, and the physical motivation behind asserting that $SE = SE\_ref + a/TI - 1/4$ (where a is a constant) is not obvious. The equation is in any case unclear, as $TI(k)$ looks like a function, but appears to be a distribution, from the description on line 24. Section 2.1, page 4, lines 26-28: The method by which the $k\_TI$ values in table 4 have been derived should be explained, to aid reproducibility.

Section 3.1, page 8, line 3-4: what does it mean, to say that the mean is below 1%? Section 3.1, page 8, line 10: "converge" should read "convergence". Section 3.2, page 11, Figure 5: This figure is difficult to understand. Does the y-axis label "difference in" refer to a change between adjacent function evaluations? What is the definition of "potential inexactness" that the grey band is representing, and what information does it give the reader about the other lines on the graph? Finally, the legend says "1.1k MC" whereas the rest of the text indicates 1200 evaluations. Section 3.3, page 12, Figure 6: More explanation is required concerning the pdf values being shown - how should they be interpreted? They are different to the pdf values being shown in Fig 5. The pdf values are presumably also not conditional on $k\_TI=1$, since they do not appear to integrate to 1? Finally, the second graph on the top line has a typo in the title: "MDT" should read "MTD". Section 3.3, page 12, line 7: Isn't the low probability of occurrence of ESD=0 and $k\_AF=0$ an input assumption? Perhaps when the meaning of the pdf

plots is more fully explained, this will become clear. Section 3.3, page 13, line 5: The "largest probability" implies total probability greater than 50% of lying within +/- 1% of the mean?

Page 13: Mostly these conclusions are justified and well-written. However, some more discussion could be given to the relative influence on the qualitative or quantitative results (i.e. differences with a deterministic approach) of the method itself, versus the specific numerical assumptions made about input parameter values, distributions and covariances.

---

## Referee Comment (RC2) · Anonymous Referee #2 · 20 Mar 2019

*Discussion of:*

**Performance of non-intrusive uncertainty quantification in the aeroservoelastic simulation of wind turbines**

19 March 2019

The authors present the application of two non-intrusive uncertainty propagation techniques: Universal Kriging and Polynomial Chaos Expansion, as means of propagating the effect of uncertainty in wind conditions and blade aerodynamics on wind turbine loads. The manuscript describes the process of setting up the uncertainty propagation models and demonstrates an application on a 10MW research turbine. In the results section, the authors show how the uncertainty in two variables – the airfoil unevenness, and the extent of degradation along the blade span, affect the distribution of various wind turbine load components. The article is well structured and clearly written, and deals with a relevant scientific problem. In my opinion, the manuscript will benefit scientifically if the authors go in further depth in some aspects of their analysis. These recommendations are given in the comments below.

**General comments**

1) In several places in the paper (e.g. page 5, line 3) the authors state that there are some potentially significant sources of uncertainty, which are not considered in order to allow more focus on other relevant uncertainty sources. This is reasonable; however in such a situation it is important to understand what is the effect of not considering these uncertainties. For example, would the ignored uncertainties have the same effect over the entire variable space considered, meaning that they will not mask the relative effects of other uncertainties? Or will their effect mix with that of other uncertainties meaning a larger model error in general?

2) The uncertainty propagation models are trained based on variable spaces with beta-distributed marginal variables. Then the probability density functions for the response surfaces are plotted based on a Monte Carlo simulation which apparently uses the abovementioned marginal distributions. However, these sampling distributions do not fully correspond to the real-world distributions of the uncertainty variables. It is therefore difficult to judge on whether a given load event is critical as it may have a high probability of occurrence in the sampling space used to train the uncertainty propagation model, but low probability in the real world, and vice versa. I suggest that the authors redo the MC analysis (Figure 6) using realistic joint distributions of the uncertainty variables. This is also a key distinguishing point between uncertainty propagation and uncertainty quantification: the response surface only propagates the uncertainty, so in order to quantify the uncertainty of the dependent variable we need to feed the propagation model with the right input uncertainties.

3) To me, the authors are considering a manifold of four random quantities: two uncertainty variables ($k_{AF}$ and $ESD$) combined with two environmental conditions – wind speed, and turbulence intensity (and wind shear as fully dependent on the latter two). I think it will make the paper clearer if the presentation is made along this logic. In this way one can also distinguish between point-to-point uncertainty between individual realizations, and the effect of the two uncertainty

factors integrated over the joint distribution of the environmental conditions (which is what I believe is the purpose of Figure 6 in the current manuscript).

4) It is not clear whether the results reported in Figure 6 are averaged over the wind speed or not. If we were considering integrated quantities such as e.g. fatigue loads, it would be relevant to show the average values. However, when talking about extremes it would be more appropriate to not do any averaging, and instead include the wind speed as one of the factors in computing the pdf of the extreme loads. This also relates to the comments above.

**Specific comments**:

5) Page 3, line 20 (first paragraph of Section 2): This is a classification of the uncertainties according to the physical mechanism that causes them. Another maybe even more relevant classification could be according to their type, e.g., statistical, measurement, model, human-caused… This should make it easier to categorize the uncertainties.

6) Page 3, lines 23-25: "Not only the nominal values of all such parameters are uncertain, but additional sources of uncertainty are introduced by manufacturing processes and the status of wear and tear of each individual machine or component". Another uncertainty source which the authors should consider here is the measurement uncertainty: the observed value of a given variable is different from its true value due to imperfect observation. This also means that we don't necessarily know the true reference.

7) Page 4, line 8: The authors describe that turbulence boxes include random realizations of a turbulence field. It would be useful to describe in more details what are the statistical properties of these randomly generated fields – e.g. are they Gaussian, what are the spectral parameters.

8) Page 4, line 12: "…These effects may alter in a significant way the statistics of the wind at a given site. All such effects are difficult to measure and quantify with precision…" What the authors refer to may be considered as a kind of measurement (epistemic) uncertainty due to not being able to quantify the variables with sufficient precision. A specific reference to this type of uncertainty can be found in Tarp-Johansen at al. [1] where this is referred to as "Exposure uncertainty".

9) Page 4, eq. 1: Please note that in Dimitrov et al. (2015) the reference turbulence intensity $TI_{ref}$ is a function of the turbulence quantile, i.e., the wind shear distribution changes with respect to the turbulence quantile. What kind of turbulence quantile have the authors considered as $TI_{ref}$? Is that taken into account by the uncertainty factor $k_{TI}$? I think the authors have to explain the relationship between the turbulence quantile and $k_{TI}$.

10) Page 5, line 14: "either uniform or a beta probability distribution" – why either distributions and not one specific?

11) Page 6, line 22: Is the severity of surface degradation $k_{AF}$ assumed to be uniform over the full extent of spanwise degradation (ESD)? I would suggest that a more realistic approach would be to have 1) $k_{AF}$ as a random, spatially-correlated variable over the blade span, and 2) the expected value of $k_{AF}$ to gradually increase towards the blade tip. This could still amount to some integrated degradation measure.

12) Page 6, line 9: What was the trend function used in the Universal Kriging approach? One could consider e.g. a polynomial chaos expansion as a trend function – one could even make use of the NIPCE already trained as a standalone model.

13) Page 6, line 29 (and Figure 3): what turbulence quantile does $TI_{ref}$ refer to? Why is the turbulence uncertainty factor $k_{TI}$ beta-distributed, normally one could use the standard assumption that the turbulence (standard deviation of wind speed) is log-normally distributed? Again, in continuation to a previous comment, we need an explanation of the relationship between the turbulence probability distribution and the uncertainty factor $k_{TI}$ and what are the implications of replacing the turbulence distribution with $k_{TI}$.

14) Page 9, Table 4: are these statistics based on the full data set over all wind speeds? Have the results been Weibull-weighted according to a certain wind speed probability, or is the wind speed probability considered uniform? Is the "standard deviation" the sample standard deviation, or the uncertainty in the mean estimate?

15) Page 10, line 7: give a definition of the collocation ratio

16) Page 10, line 8: what is the sampling distribution of the MC? Is that the same as the MC sample used to train the models?

17) Page 10, line 11: the UK converges faster than the NIPCE. Could that be because Kriging is in essence an interpolation scheme, and the response is linear enough (as the authors point out themselves) so that a few points are sufficient to establish a reasonable extrapolation?

18) Figure6: there are some "wrinkles" in the contour plots. Could these be caused by having few data points (40 function evaluations only)? What if we added more data – maybe the contours would resemble more straight lines (= closer to linear dependencies)?

19) Page 13, line 24: "…the deterministic conditions prescribed by international design standards generate maximum values of loads and power production, which however are typically associated with a very low probability of occurrence". This is guaranteed only if the sampling distribution used to propagate the uncertainty is the same as the real-world distribution of the random input variables. As discussed in the general comments, this is not necessarily the case with the present data sets.

**Technical comments**:

20) Page 4, line 14: "give turbine" -> "given turbine"
21) Page 8, line 10: "converge" -> "convergence"

**References**:

[1] Tarp-Johansen, N., Madsen, H. & Frandsen, S. T. (2002) Partial safety factors for extreme load effects. *Technical report Risø-R-1319(EN)*, Risoe National Laboratory, Denmark

---

## Author Comment (AC1) · 23 May 2019

**REVISION TO MANUSCRIPT DRAFT**

**Wind Energy Science Discussion**

**Performance of non-intrusive uncertainty quantification in the aeroservoelastic simulation of wind turbines**

The authors would like to thank the two reviewers for their time and for the useful feedback. All inputs that they provided have contributed to the improvement of the paper.

A list of point-by-point replies to the reviewers' comments is reported in the following.

**Reviewer #1**

**Numbered comments**

1. [**Reviewer**] *The authors present a well-written and well-motivated application of standard non-intrusive uncertainty calculations to the estimation of loads on a wind turbine. There are, however, a few areas where additional clarity or corrections to the text and figures are required. Abstract: In the first line, "uncertainties" should be replaced by "aleatory uncertainties".*

   [**Authors**] The word "aleatory" has been added, as suggested.

2. [**Reviewer**] *The last sentence should also be made more specific as to what the effects and shortcomings are.*

   [**Authors**] The whole abstract has been reformulated and made more specific.

3. [**Reviewer**] *Section 2, page 1, line 21: "uncertainties are [...] only indirectly accounted for" – the concept of "indirect" uncertainty calculation should be explained, preferably with a citation of an example.*

   [**Authors**] An example has been added to the text.

4. [**Reviewer**] *Section 2.1, page 4, line 14: "give" should read "given".*

   [**Authors**] The typo has been corrected.

5. [**Reviewer**] *Section 2.1, page 4, lines 16-20: The choice of a Beta distribution (actually strictly speaking a scaled Beta distribution, since the input values do not always lie between 0 and 1 – see also Section 2.2, page 5, line 21 for ESD) is not sufficiently motivated. This distribution has some specific purposes in the statistical literature, in particular for expressing an uncertainty distribution over a probability. The reason for the turbulence intensity to be modified by a factor which lies between 0.5 and 2 is not explained, since it implies that the turbulence intensity corresponding to $k\_TI=1$ will not actually be the mean or median of this distribution? A log-normal or truncated Gaussian distribution (with mean or median set to 1) would appear more appropriate.*

   [**Authors**] The focus of the present study is on uncertainty propagation methods that are applicable to wind energy problems and that can converge faster than the standard

Monte Carlo (MC) approach. To perform the necessary comparisons, realistic uncertainties were generated from proprietary datasets. It was observed that the distributions of these datasets could be accurately modeled by scaled beta distributions. In the datasets, TI was not distributed symmetrically, and this is why $k\_TI=1$ is not centered around 1.

It should however be highlighted that neither NIPCE nor Kriging are bound to (scaled) Beta distributions and that other distributions could be readily used. Log-normal or truncated Gaussian distributions would be a perfectly feasible option. This said, in this study the scaled beta distributions nicely met our needs, i.e. representing data that is bound and not necessary symmetrically distributed.

The text in the manuscript has been changed to better explain these points.

6. [**Reviewer**] *Section 2.1, page 4, lines 21-22: The Dimitrov paper does not appear to contain this equation, and the physical motivation behind asserting that SE = SE_ref + a/TI - 1/4 (where a is a constant) is not obvious. The equation is in any case unclear, as TI(k) looks like a function, but appears to be a distribution, from the description on line 24.*

[**Authors**] The equation is number five in the paper from Dimitrov et al., 2015 (https://doi.org/10.1002/we.1797). The difference between the equation of Dimitrov et al., 2015 and our equation is that in our work TI is not only dependent on wind speed, but also on the uncertain parameter $k_{TI}$. For varying values of $k_{TI}$, TI changes and so does the shear coefficient.

We rewrote a large portion of the section to better explain this point.

7. [**Reviewer**] *Section 2.1, page 4, lines 26-28: The method by which the k_TI values in table 4 have been derived should be explained, to aid reproducibility.*

[**Authors**] The focus of the paper is on uncertainty propagation and results can be reproduced by using the parameters reported in Table 3. These values are site and wind turbine dependent, and different values would certainly change the outputs, without however invalidating the methods used for uncertainty propagation.

Lines 26-28 and the corresponding references were removed in the new version of the manuscript, as they did not help with the understanding and were therefore deemed to be superfluous.

8. [**Reviewer**] *Section 3.1, page 8, line 3-4: what does it mean, to say that the mean is below 1%?*

[**Authors**] The sentence was imprecise, and it has been improved. The sentence refers to the convergence trends and to the variations in mean and standard deviation over the iterations.

9. [**Reviewer**] *Section 3.1, page 8, line 10: "converge" should read "convergence".*

[**Authors**] This typo has now been corrected.

10. *[**Reviewer**] Section 3.2, page 11, Figure 5: This figure is difficult to understand. Does the y-axis label "difference in" refer to a change between adjacent function evaluations? What is the definition of "potential inexactness" that the grey band is representing, and what information does it give the reader about the other lines on the graph? Finally, the legend says "1.1k MC" whereas the rest of the text indicates 1200 evaluations.*

    [**Authors**] The y-axis represents the difference with respect to the MC estimates obtained with 1,100 sample points. As the legend of Figure 5 reports, "The gray area reflects the potential inexactness of the MC benchmark, and it represents the 95% confidence intervals for 1,100 sampling points.". "Potential inexactness" then accounts for the fact that, with a finite number of sampling points (here, equal to 1,100), MC estimates the outputs only up to some possible residual variations. The grey band could be made narrower by increasing the number of samples. The text was updated to clarify this point. The number 1,200 referred to older calculations, while 1,100 is the correct number. The text was corrected accordingly.

11. *[**Reviewer**] Section 3.3, page 12, Figure 6: More explanation is required concerning the pdf values being shown - how should they be interpreted? They are different to the pdf values being shown in Fig 5. The pdf values are presumably also not conditional on k_TI=1, since they do not appear to integrate to 1? Finally, the second graph on the top line has a typo in the title: "MDT" should read "MTD".*

    [**Authors**] Figure 6 shows the values and corresponding probabilities of each key output for combinations of $k_{AF}$ and ESD. These values correspond to a 2D slice of the tri-dimensional space. The slice is cut for $k_{TI}=1$.

    Two plots are defined for each key output: on the left a plot shows the percent difference with respect to the mean of the various key outputs for the different possible input combinations of $k_{AF}$ and ESD, while the plot on the right shows the corresponding probabilities. The plots were generated by evaluating the UK model, trained with 40 function evaluations, with a large random sample of 1,000,000 points, using $k_{TI}=1$. The probabilities were then computed using this sample, so they are conditional on $k_{TI}=1$.

    The pdf shown in Fig. 4 (not Fig. 5, which has no pdfs) corresponds to the sample of 1,100 points obtained from Monte Carlo simulations.

    The typo in the title of the graph has now been corrected and a more complete description of Fig. 6 has been added at the beginning of Sect. 3.3.

12. *[**Reviewer**] Section 3.3, page 12, line 7: Isn't the low probability of occurrence of ESD=0 and k_AF=0 an input assumption? Perhaps when the meaning of the pdf plots is more fully explained, this will become clear.*

    [**Authors**] Yes, it is. The text has been changed to clarify this point.

13. *[**Reviewer**] Section 3.3, page 13, line 5: The "largest probability" implies total probability greater than 50% of lying within +/- 1% of the mean?*

[**Authors**] The sentence in the text has now been reformulated to highlight that the highest probabilities of occurrence correspond to values of MTD that fall very close to the mean values, and that the deterministic condition prescribed by the standards actually corresponds to the lowest probability of occurrence.

14. [***Reviewer***] *Page 13: Mostly these conclusions are justified and well-written. However, some more discussion could be given to the relative influence on the qualitative or quantitative results (i.e. differences with a deterministic approach) of the method itself, versus the specific numerical assumptions made about input parameter values, distributions and covariances*

    [**Authors**] We thank the reviewer for the useful comments and suggestions. We hope that our changes improved the text. We are aware that this work represents only a preliminary step and much remains to be done before these methods for uncertainty propagation are fully understood and become widely applicable. A better analysis of the outputs is a top priority, especially to evaluate the impact of these methods on design. In fact, work is ongoing to integrate the UQ approach within a design framework. A sentence on future work was added to this section, to highlight this point.

**Reviewer #2**

*The authors present the application of two non-intrusive uncertainty propagation techniques: Universal Kriging and Polynomial Chaos Expansion, as means of propagating the effect of uncertainty in wind conditions and blade aerodynamics on wind turbine loads. The manuscript describes the process of setting up the uncertainty propagation models and demonstrates an application on a 10MW research turbine. In the results section, the authors show how the uncertainty in two variables – the airfoil unevenness, and the extent of degradation along the blade span, affect the distribution of various wind turbine load components. The article is well structured and clearly written, and deals with a relevant scientific problem. In my opinion, the manuscript will benefit scientifically if the authors go in further depth in some aspects of their analysis. These recommendations are given in the comments below.*

***General comments***

1. [***Reviewer***]: *In several places in the paper (e.g. page 5, line 3) the authors state that there are some potentially significant sources of uncertainty, which are not considered in order to allow more focus on other relevant uncertainty sources. This is reasonable; however in such a situation it is important to understand what is the effect of not considering these uncertainties. For example, would the ignored uncertainties have the same effect over the entire variable space considered, meaning that they will not mask the relative effects of other uncertainties? Or will their effect mix with that of other uncertainties meaning a larger model error in general?*

    [**Authors**] This is a very good point, which however we have not yet addressed and that -to be fully answered- indeed requires methods like the ones presented in this work. The

problem of uncertainty quantification in wind turbine simulation and design is very complex due to multiple reasons. One of them is exactly the one raised by the reviewer: what are the most "important" uncertainties? Answering this question with a standard MC approach is extremely expensive, to the point of being undoable. To address this problem, we started by testing different uncertainty propagation methods, in order to identify the most suitable one. To run the necessary comparisons, a sub-set of uncertainties that we could quantify was selected. We agree with the reviewer that the next natural step is a detailed assessment of the importance of all uncertainties impacting wind turbine analysis and design.

These thoughts were already included in the outlook for future work, but we have now added one additional sentence to better elaborate on them. In addition, although the introduction already clearly stated the goals of this paper, we have now added a new sentence that clarifies that an in-depth study of the effects of uncertainties is not one of them.

2.  *[**Reviewer**] The uncertainty propagation models are trained based on variable spaces with beta-distributed marginal variables. Then the probability density functions for the response surfaces are plotted based on a Monte Carlo simulation which apparently uses the abovementioned marginal distributions. However, these sampling distributions do not fully correspond to the real-world distributions of the uncertainty variables. It is therefore difficult to judge on whether a given load event is critical as it may have a high probability of occurrence in the sampling space used to train the uncertainty propagation model, but low probability in the real world, and vice versa. I suggest that the authors redo the MC analysis (Figure 6) using realistic joint distributions of the uncertainty variables. This is also a key distinguishing point between uncertainty propagation and uncertainty quantification: the response surface only propagates the uncertainty, so in order to quantify the uncertainty of the dependent variable we need to feed the propagation model with the right input uncertainties.*

    [**Authors**] This a second very good point raised by the reviewer. However, as clearly stated throughout the text, this work limits its scope to the testing of two propagation methods, analyzing their convergence trends and performing an initial analysis of the uncertain outputs. We did not (and still do not) have access to distributions of the uncertain inputs coming from the real-world. These data sets would be extremely valuable, also to address Comment #1. This work aims at showing that NIPCE and Universal Kriging are two valuable alternatives to MC for the propagation of uncertainties affecting wind turbines. A second goal of this work is to show that the world of UQ has a very large potential to better estimate outputs of interest and help reducing safety factors in wind turbine design.

3.  *[**Reviewer**] To me, the authors are considering a manifold of four random quantities: two uncertainty variables ($k_{AF}$ and $ESD$) combined with two environmental conditions – wind speed, and turbulence intensity (and wind shear as fully dependent on the latter two). I think it will make the paper clearer if the presentation is made along this logic. In this way*

*one can also distinguish between point-to-point uncertainty between individual realizations, and the effect of the two uncertainty factors integrated over the joint distribution of the environmental conditions (which is what I believe is the purpose of Figure 6 in the current manuscript).*

[**Authors**] The work adopts three random quantities, $k_{AF}$, ESD and $k_{TI}$. Shear is linked to TI through Eq. 1. Wind speed is not an uncertain parameter and simulations are run for wind speed bins of 2 m/s from cut in to cut out. We thought of adopting the logic proposed by the reviewer of analyzing the single uncertainties. However, no strong conclusion could be drawn by that approach, and we finally opted for a more aggregated analysis of the results.

4. [***Reviewer***] *It is not clear whether the results reported in Figure 6 are averaged over the wind speed or not. If we were considering integrated quantities such as e.g. fatigue loads, it would be relevant to show the average values. However, when talking about extremes it would be more appropriate to not do any averaging, and instead include the wind speed as one of the factors in computing the pdf of the extreme loads. This also relates to the comments above.*

[**Authors**] The results are not averaged over the wind speed. Quantities such as MTD, ThS, CBRM and CTBM are computed by taking the maximum values across all wind speeds. Quantities such as AEP and the three DELs are instead integrated across the wind speeds assuming the Weibull distribution corresponding to Class IA (k=2, $U_{avg}$=10).

We added a paragraph in Sect. 3.1 to better explain this point.

**Specific comments**

5. [***Reviewer***] *Page 3, line 20 (first paragraph of Section 2): This is a classification of the uncertainties according to the physical mechanism that causes them. Another maybe even more relevant classification could be according to their type, e.g., statistical, measurement, model, human-caused... This should make it easier to categorize the uncertainties.*

[**Authors**] Following the reply to Comment #2, the focus of the present work is to test uncertainty propagation methods for three realistic uncertain inputs. An important, but also very challenging, work would be to categorize the input uncertainties and assess their importance. This would be very valuable to the scientific community, although the lack of measurements and field data available in the public domain complicates this task. Although very useful, we believe this aspect to be outside of the scope of the present work.

6. [***Reviewer***] *Page 3, lines 23-25: "Not only the nominal values of all such parameters are uncertain, but additional sources of uncertainty are introduced by manufacturing processes and the status of wear and tear of each individual machine or component". Another uncertainty source which the authors should consider here is the measurement*

*uncertainty: the observed value of a given variable is different from its true value due to imperfect observation. This also means that we don't necessarily know the true reference.*

[**Authors**] We agree with the reviewer on this point, and we added this source of uncertainty in the text.

7. [**Reviewer**] *Page 4, line 8: The authors describe that turbulence boxes include random realizations of a turbulence field. It would be useful to describe in more details what are the statistical properties of these randomly generated fields – e.g. are they Gaussian, what are the spectral parameters.*

   [**Authors**] Turbulence fields were generated adopting the standard values prescribed by IEC standards. Only TI and shear exponent were assumed uncertain and perturbed.

8. [**Reviewer**] *Page 4, line 12: "…These effects may alter in a significant way the statistics of the wind at a given site. All such effects are difficult to measure and quantify with precision…" What the authors refer to may be considered as a kind of measurement (epistemic) uncertainty due to not being able to quantify the variables with sufficient precision. A specific reference to this type of uncertainty can be found in Tarp-Johansen at al. [1] where this is referred to as "Exposure uncertainty".*

   [**Authors**] When modeling the wind, the distinction between aleatory and epistemic uncertainties may blur. In this work we addressed the first ones, but it is however true that wind is also affected by epistemic uncertainties that should be addressed. We thank the reviewer for having provided a reference we were not aware of. This reference has now been included in the revised version of the manuscript.

9. [**Reviewer**] *Page 4, eq. 1: Please note that in Dimitrov et al. (2015) the reference turbulence intensity $TI_{ref}$ is a function of the turbulence quantile, i.e., the wind shear distribution changes with respect to the turbulence quantile. What kind of turbulence quantile have the authors considered as $TI_{ref}$? Is that taken into account by the uncertainty factor $k_{TI}$? I think the authors have to explain the relationship between the turbulence quantile and $k_{TI}$.*

   [**Authors**] Following Comment #6 of Reviewer #1, the paragraph has been reformulated. The distribution shown in Fig. 3 was determined for a turbulence quantile of 90%.

   [**Reviewer**] *Page 5, line 14: "either uniform or a beta probability distribution" – why either distributions and not one specific?*

   [**Authors**] This was a typo and we corrected it. In the preliminary studies we did not have any indication on $k_{AF}$ and we therefore ran the first analyses adopting a uniform distribution. Later on during the study, one of the authors gained access to real data and provided the values of α and β reported in Table 3.

   [**Reviewer**] *Page 6, line 22: Is the severity of surface degradation $k_{AF}$ assumed to be uniform over the full extent of spanwise degradation (ESD)? I would suggest that a more realistic approach would be to have 1) $k_{AF}$ as a random, spatially-correlated variable over*

*the blade span, and 2) the expected value of $k_{AF}$ to gradually increase towards the blade tip. This could still amount to some integrated degradation measure.*

[**Authors**] We agree with Reviewer #2 that this could be a better approach for future studies. Nonetheless, the (few) experimental data points used to fill Table 3 suggested a constant $k_{AF}$.

10. [***Reviewer***] *Page 6, line 9: What was the trend function used in the Universal Kriging approach? One could consider e.g. a polynomial chaos expansion as a trend function – one could even make use of the NIPCE already trained as a standalone model.*

[**Authors**] The trend function used in the UK approach is a reduced quadratic polynomial.

11. [***Reviewer***] *Page 6, line 29 (and Figure 3): what turbulence quantile does $TI_{ref}$ refer to? Why is the turbulence uncertainty factor $k_{TI}$ beta-distributed, normally one could use the standard assumption that the turbulence (standard deviation of wind speed) is log-normally distributed? Again, in continuation to a previous comment, we need an explanation of the relationship between the turbulence probability distribution and the uncertainty factor $k_{TI}$ and what are the implications of replacing the turbulence distribution with $k_{TI}$.*

[**Authors**] See Comment #9, the whole paragraph has been reformulated to better explain how $k_{TI}$ was defined.

12. [***Reviewer***] *Page 9, Table 4: are these statistics based on the full data set over all wind speeds? Have the results been Weibull-weighted according to a certain wind speed probability, or is the wind speed probability considered uniform? Is the "standard deviation" the sample standard deviation, or the uncertainty in the mean estimate?*

[**Authors**] These statistics are based on a sample of 1,100 MC function evaluations. Each function evaluation corresponds to 12 transient simulations at different wind speeds from cut-in to cut-out, considering six turbulent seeds. The extreme loads (MTD, ThS, CBRM, CTBM) are computed extracting the maximum overall value of each simulation for each quantity. The DELs (DEL ThS, DEL CBRM, DEL CTBM) and AEP are computed for each dynamic simulation and Weibull-averaged according to the Weibull of wind class 1A. The standard deviation is computed as the amount of dispersion of the key outputs of the 1,100 function evaluations.

Text has been changed to include the above information.

13. [***Reviewer***] *Page 10, line 7: give a definition of the collocation ratio*

[**Authors**] The collocation ratio is defined as the ratio between the number of function evaluations used to train the model and the total number of terms in the chaos expansion.

The definition has now been added to the text.

14. [***Reviewer***] *Page 10, line 8: what is the sampling distribution of the MC? Is that the same as the MC sample used to train the models?*

[**Authors**] The sampling distribution of the MC is random. Yes, it is the same used to train the models.

15. [**Reviewer**] *Page 10, line 11: the UK converges faster than the NIPCE. Could that be because Kriging is in essence an interpolation scheme, and the response is linear enough (as the authors point out themselves) so that a few points are sufficient to establish a reasonable extrapolation?*

    [**Authors**] Yes, we believe this to be a correct interpretation of the results.

16. [**Reviewer**] *Figure 6: there are some "wrinkles" in the contour plots. Could these be caused by having few data points (40 function evaluations only)? What if we added more data – maybe the contours would resemble more straight lines (= closer to linear dependencies)?*

    [**Authors**] The contour plots in Fig. 6 are computed by evaluating 1,000,000 random sample points in the UK model trained with 40 function evaluations. We did ask ourselves the same question at the time of analyzing the outputs of the simulations. We then generated similar contour plots training the model with more evaluation points. However, the plots did not change substantially, and we therefore concluded that the wrinkles are likely associated to non-linearities of the aeroservoelastic model.

17. [**Reviewer**] *Page 13, line 24: "…the deterministic conditions prescribed by international design standards generate maximum values of loads and power production, which however are typically associated with a very low probability of occurrence". This is guaranteed only if the sampling distribution used to propagate the uncertainty is the same as the real-world distribution of the random input variables. As discussed in the general comments, this is not necessarily the case with the present data sets.*

    [**Authors**] The uncertain input parameters reported in Table 3 come from real datasets and can be assumed to be realistic. It is true that a higher number of uncertain parameters will likely increase the uncertainty of the outputs. Nonetheless, the variations observed in this work suggest that the adoption of uncertainty propagation methods may help reducing safety factors, possibly drastically.

**Technical comments:**

18. [**Reviewer**] *Page 4, line 14: "give turbine" -> "given turbine"*

    [**Authors**] The typo has been corrected.

19. [**Reviewer**] *Page 8, line 10: "converge" -> "convergence"*

    [**Authors**] The typo has been corrected.

We have taken the opportunity to make several small editorial changes to the text, in order to improve readability. A revised version of the manuscript is attached to the present reply, with the main changes highlighted in blue.

The authors